# Clinical Discernment, Bone Marrow, and Molecular Diagnostics Are Equally Important to Solve the Phenotypic Mimicry among Subtypes of Myeloproliferative Neoplasms

Susann Schulze [1], Nadia Jaekel [2], Christin Le Hoa Naumann [2], Anja Haak [3], Marcus Bauer [3], Claudia Wickenhauser [3] and Haifa Kathrin Al-Ali [1,2,*]

1 Krukenberg-Cancer Center Halle, University Hospital Halle, 06120 Halle (Saale), Germany; susann.schulze2@uk-halle.de
2 Department of Haematology/Oncology, University Hospital Halle, 06120 Halle (Saale), Germany; nadja.jaekel@uk-halle.de (N.J.); christin.naumann@uk-halle.de (C.L.H.N.)
3 Institute of Pathology, University Hospital of Halle, 06112 Halle (Saale), Germany; anja.haak@uk-halle.de (A.H.); marcus.bauer@uk-halle.de (M.B.); claudia.wickenhauser@uk-halle.de (C.W.)
* Correspondence: haifa.al-ali@uk-halle.de; Tel.: +49-345-557-7712; Fax: +49-345-557-7720

**Abstract:** The 2016 WHO classification integrates clinical, bone marrow (BM)-morphology, and molecular features to define disease entities. This together with the advancements in molecular detection and standardization of BM features enable an accurate diagnosis of myeloproliferative neoplasms (MPN) in the majority of patients. Diagnostic challenges remain due to phenotypic mimicry of MPN, failing specificity of BM-morphology, and the fact that phenotype-driver mutations, such as *JAK2*V617F, are not exclusive to a particular MPN, and their absence does not preclude any of these. We present a series of cases to illustrate themes to be considered in complex cases of MPN, such as triple-negative (TN)-MPN or MPN-unclassifiable (MPN-U). Eleven patients labelled as TN-MPN or MPN-U were included. Serum tryptase and NGS were part of a systematic/sequential multidisciplinary evaluation. Results were clustered into four categories based on diagnostic entities and/or how these diagnoses were made: (A) With expanding molecular techniques, *BCR-ABL1* and karyotyping should not be missed; (B) systemic mastocytosis is underdiagnosed and often missed; (C) benign non-clonal disorders could mimic MPN; and (D) NGS could prove clonality in some "TN"-MPN cases. The prognostic/therapeutic consequences of an accurate diagnosis are immense. In TN-MPN or MPN-U cases, a multidisciplinary re-evaluation integrating molecular results, BM-morphology, and clinical judgment is crucial.

**Keywords:** triple-negative myeloproliferative neoplasms; MPN-unclassifiable; mastocytosis; molecular diagnostics; next-generation sequencing

## 1. Introduction

In the 2016 edition of the World Health Organization (WHO) classification system of tumours of the hematopoietic and lymphoid tissues, chronic myeloid neoplasms are classified into myelodysplastic syndromes (MDS), myeloproliferative neoplasms (MPN), MDS/MPN overlap, and myeloid/lymphoid neoplasms with eosinophilia and recurrent rearrangements of *PDGFRA*, *PDGFRB*, and *FGFR1* or *PMC1-JAK2*. The category of MPN includes the three major *JAK2/CALR/MPL* mutation-related MPNs (i.e., polycythaemia vera (PV), essential thrombocythemia (ET), and primary myelofibrosis (PMF)) as well as four other clinicopathologic entities: chronic myeloid leukaemia (CML), chronic neutrophilic leukaemia (CNL), chronic eosinophilic leukaemia, and not otherwise specified (CEL-NOS) and MPN unclassifiable (MPN-U). The latter subgroup includes MPN-like neoplasms that cannot be clearly classified as one of the other six subcategories of MPNs. Of note, mastocytosis is no longer classified under the MPN category [1]. Integrating clinical, bone marrow

(BM) findings, and molecular features is the most suitable attempt to define disease entities. This together with the advancements regarding the standardization of morphological bone marrow (BM) features enables an accurate diagnosis and differentiation of MPN subtypes in the majority of patients.

Yet, in routine clinical practice, diagnostic challenges remain. First, a multidisciplinary discussion at the highest level is a prerequisite for managing the phenotypic mimicry among MPNs with other myeloid neoplasms and even benign hematopoietic disorders [2]. Second, there is the issue of the failing specificity of BM morphology for differentiation of some MPN entities and their separation from MDS/MPN overlap syndromes or MDS-particularly in clonally undefined PMF [3,4]. Indeed, the reproducibility of the BM characteristics as described in the WHO classification remains a debate issue [2]. Thirdly, phenotype-driver mutations, such as *JAK2*, *CALR*, and *MPL* mutations, are not mutually exclusive and not exclusive to a particular MPN and their absence does not preclude any of these neoplasms. A positive mutation assay establishes the presence of a clonal haematopoiesis and not its identity, and the absence of a specific mutation does not rule out MPN. In fact, up to 20% of ET and 10–15% of PMF patients have no canonical *JAK2*, *MPL*, or *CALR* driver mutations and are currently referred to as "triple-negative" (TN) [5,6]. Despite the above-mentioned challenges, it needs to be remembered that the diagnosis of MPN is a multidisciplinary task requiring consideration of the presenting clinical features, morphological assessment of the peripheral blood and bone marrow aspirate (cytology) and biopsy, standard laboratory parameters, and, ever increasingly, the underlying acquired genetic status. Actually, the treating physician holds the reins, as all diagnostic pillars are available to her/him. An accurate diagnosis distinguishing between the different subtypes of MPN and their separation from MDS/MPN and systemic mastocytosis (SM) is of utmost clinical importance, as treatment options and outcome for the different subtypes vary significantly.

In this work, a cohort of clinical cases are presented to underscore essential points that need to be integrated in the diagnostic work-up when faced with complex cases of MPN, such as ill-defined TN- and/or treatment-refractory MPN or MPN-U.

## 2. Patients and Methods

A cohort of eleven patients referred to the University Hospital Halle in 2019 because of treatment-refractory TN-MPN or clonally ill-defined MPN-U is included. Canonical mutations in the *JAK2*, *CALR*, and *MPL* genes by standard polymerase chain reaction (PCR) were already excluded by the treating physicians. Systematic re-evaluation consisted of:

A. A thorough history and clinical examination, blood picture with a differential count, and blood chemistry (liver and renal function tests as well as LDH);
B. A serum tryptase level, which is part of our routine diagnostic work-up for hematologic diseases (reference range < 11 µg/L). Further laboratory tests were done as indicated;
C. The presence of a molecular analysis for *BCR-ABL1* and classical karyotyping was checked; and
D. Fresh BM aspirates for cytology and biopsies were re-evaluated by the haematologist and pathologist at our institution, respectively.

If a definite MPN entity could not yet be clearly defined:

A. Molecular testing for *PDGFRA*, *PDGFRB*, *FGFR1*, and *PCM1-JAK2*—even if eosinophilia is not present—was conducted [1,7];
B. Deep sequencing by next-generation sequencing (NGS) was done. Currently, the molecular Panel of NEO New Oncology GmbH, Cologne, Germany is used by our institution (Figure 1); and
C. Finally, cases were presented and discussed together with haemato- and molecular-pathologists in the multidisciplinary Molecular Tumour Board of the Krukenberg Cancer Center of the University Hospital Halle.

## A) Coverage of exome hotspot regions

| Gene Name | Chromosome | Start | End | Coverage | Reads | Region |
|---|---|---|---|---|---|---|
| ABL1 | chr9 | 133738226 | 133738465 | 1901 | 5704 | Exonic_hotspot |
| ABL1 | chr9 | 133750226 | 133750466 | 2245 | 7328 | Exonic_hotspot |
| ABL1 | chr9 | 133748215 | 133748455 | 709 | 2248 | Exonic_hotspot |
| ABL1 | chr9 | 133747437 | 133747677 | 553 | 1783 | Exonic_hotspot |
| CSF3R | chr1 | 36933372 | 36933612 | 4261 | 13930 | Exonic_hotspot |
| ETNK1 | chr12 | 22811897 | 22812137 | 1179 | 3562 | Exonic_hotspot |
| FLT3 | chr13 | 28592544 | 28592784 | 2299 | 7717 | Exonic_hotspot |
| IDH1 | chr2 | 209113078 | 209113198 | 3047 | 6564 | Exonic_hotspot |
| IDH2 | chr15 | 90631849 | 90632078 | 1611 | 4075 | Exonic/UTR_hotspot |
| IDH2 | chr15 | 90631728 | 90631848 | 1235 | 3016 | Exonic_hotspot |
| JAK2 | chr9 | 5069831 | 5069951 | 782 | 1748 | Exonic_hotspot |
| JAK2 | chr9 | 5073621 | 5073861 | 1377 | 4547 | Exonic_hotspot |
| JAK2 | chr9 | 5069952 | 5070150 | 899 | 1791 | Exonic_hotspot |
| KIT | chr4 | 55599176 | 55599416 | 2166 | 7335 | Exonic_hotspot |
| KRAS | chr12 | 25398148 | 25398388 | 1056 | 3181 | Exonic/UTR_hotspot |
| KRAS | chr12 | 25380237 | 25380396 | 1358 | 2285 | Exonic_hotspot |
| MPL | chr1 | 43814962 | 43815121 | 2693 | 4274 | Exonic_hotspot |
| MPL | chr1 | 43818296 | 43818495 | 3179 | 7349 | Exonic/UTR_hotspot |
| MPL | chr1 | 43805142 | 43805270 | 2283 | 3122 | Exonic_hotspot |
| NPM1 | chr5 | 170837449 | 170837569 | 610 | 1609 | Exonic_hotspot |
| NRAS | chr1 | 115258614 | 115258854 | 3024 | 9657 | Exonic/UTR_hotspot |
| NRAS | chr1 | 115256490 | 115256689 | 2259 | 5103 | Exonic_hotspot |
| SRSF2 | chr17 | 74732948 | 74733068 | 1652 | 3429 | Exonic/UTR_hotspot |

## B) Point mutations, small insertions, and deletions

| Gene | Investigated Exons | Transcript |
|---|---|---|
| ABL1 | 1–11 | NM_007313 |
| ASXL1 | 11, 12 | NM_015338 |
| BCOR | 2–14 | NM_001123384 |
| CALR | 8, 9 | NM_004343 |
| CBL | 8, 9 | NM_005188 |
| CSF3R | 14, 17 | NM_000760 |
| DNMT3A | 2–23 | NM_175629 |
| ETNK1 | 3 | NM_018638 |
| ETV6 | 1–8 | NM_001987 |
| EZH2 | 2–20 | NM_004456 |
| FLT3* | 14, 15, 20 | NM_004119 |
| IDH1 | 4 | NM_005896 |
| IDH2 | 4 | NM_002168 |
| JAK2 | 12, 13, 14, 15 | NM_001322194 |
| KIT* | 8, 17 | NM_000222 |
| KRAS | 2, 3 | NM_004985 |
| MLL1 (KMT2A) | 1–36 | NM_001197104 |
| MPL | 4, 10, 12 | NM_005373 |
| NPM1 | 1–11 | NM_002520 |
| NRAS | 2, 3 | NM_002524 |
| RUNX1 | 2-9 | NM_001754 |
| SETBP1 | 4 | NM_015559 |
| SF3B1 | 13, 14, 15, 16 | NM_012433 |
| SRSF2 | 1 | NM_003016 |
| STAG2 | 3–35 | NM_001042749 |
| TET2 | 3–11 | NM_001127208 |
| TP53 | 2–11 | NM_000546 |
| U2AF1 | 2, 6 | NM_001025204 |
| WT1 | 7, 9 | NM_024426 |

**Figure 1.** *Cont.*

## C) Translocations

| Gene | Investigated Exons | Investigated Introns[1,2] | Transcript |
|---|---|---|---|
| ABL1 | | 1-2 | NM_007313 |
| BCR | 8, 13-14 | 1,5,8,13,14,19 | NM_004327 |
| CBFB | | 4-5 | NM_001755 |
| ETV6 | | 4-5 | NM_001987 |
| EVI1 (MECOM) | | 3 | NM_004991 |
| FGFR1 | | 10 | NM_001174067 |
| JAK2 | 11, 12-15 | 8,10,11,16,18 | NM_001322194 |
| MLL1 (KMT2A) | | 6-14 | NM_001197104 |
| MLLT2 (AF4, AFF1) | 5 | 4 | NM_001166693 |
| MYH11 | 29-31, 33 | 28-33 | NM_001040114 |
| PCM1 | | 26, 36 | NM_001315507 |
| PDGFRA | 12 | 11 | NM_001347829 |
| PDGFRB | 2-23 | 8-11 | NM_002609 |
| PML | 6 | 3, 6 | NM_033238 |
| RARA | | 2 | NM_000964 |
| RPN1 | | 1 | NM_002950 |
| RUNX1 | | 6 | NM_001754 |
| RUNX1T1 | | 2 | NM_175634 |

**Figure 1.** The current myeloid next-generation sequencing panel. (**A**) Coverage of exome hotspot regions. (**B**) Point mutations, small insertions, and deletions. (**C**) Translocations.

## 3. Results

The diagnostic work-up of the eleven patients was clustered into four categories based on diagnostic entities and/or how these diagnoses were made. A summary of the patients, initial referral diagnoses, main clinical features, the diagnostic hint that ultimately lead to the final diagnosis, and the therapeutic options can be found in Table 1.

**Table 1.** Patients, initial referral diagnosis, diagnostic clue, final diagnosis, and treatment.

| Case | Gender | Age (y) | Referral Diagnosis | Clinical Characteristics | Somatic Gene Mutation per NGS | Diagnostic Hint | Final Diagnosis | Therapy |
|---|---|---|---|---|---|---|---|---|
| #1 | F | 65 | TN prefibrotic MF | Platelets >1 million/μL | Not done | fusion gene *BCR-ABL1* | CML | Nilotinib |
| #2 | F | 66 | TN ET | RBC-TD Platelets >1 million/μL | *BCOR* | Cytogenetics: MDS del(5q) | MDS del(5q) | Lenalidomide |
| #3 | M | 63 | TN prefibrotic MF | RBC-TD, Ascites, wasting | *cKIT*D816V | Tryptase >200 ng/mL | ASM | Midostaurin, HCT |
| #4 | F | 55 | MPN-U | RBC-TD, wasting, dysphagia (PEG) | *KRAS* *cKIT*D816V | Tryptase 49 ng/mL | ASM | Midostaurin, HCT |
| #5 | F | 52 | TN PV | Leucocytosis | *NPM1* | Tryptase >200 ng/mL | SM-AHN (AML) | Chemo-therapy, Imatinib, HCT |
| #6 | M | 63 | MDS/MPN-overlap | Pancytopenia, splenomegaly | *EZH2* | Tryptase 34 ng/mL | SM-AHN (MDS) | Midostaurin, HCT |
| #7 | F | 69 | MPN-U | Pleural effusions, splenomegaly | *NRAS* | Tryptase >200 ng/mL | SM-AHN (MPN) | Midostaurin |

<div align="center">Table 1. <em>Cont.</em></div>

| Case | Gender | Age (y) | Referral Diagnosis | Clinical Characteristics | Somatic Gene Mutation per NGS | Diagnostic Hint | Final Diagnosis | Therapy |
|------|--------|---------|--------------------|--------------------------|-------------------------------|-----------------|-----------------|---------|
| #8 | M | 31 | TN PV | retinal vein thrombosis | Not done | rare heterozygous variant in the beta-globin-chain (point mutation in exon 2 (c.119A > C) | congenital erythrocytosis | Phlebotomy, ASA |
| #9 | F | 74 | TN ET | RBC-TD, platelets >3 million/µl | None | Raynaud disease | cold agglutinin hemolytic anemia | Steroids |
| #10 | M | 56 | TN PMF | RBC-TD, wasting | *JAK2* c.3323A > G p.N1108S; (exon 25) *ASXL1 RUNX1* | sequencing | PMF | Ruxolitinib, HCT |
| #11 | F | 41 | TN MPN-U | Fatigue, splenomegalie | *JAK2* c.3188G > A, p.R1063H (exon 25) | sequencing | PMF | Ruxolitinib |

AML, acute myeloid leukaemia; ASM, aggressive systemic mastocytosis; ASA, acetylsalicylic acid; *ASXL1*, additional sex combs-like 1; *BCOR*, B-cell-lymphoma-6-co-repressor; *BCR-ABL1*, breakpoint cluster region-Abelson murine leukaemia; *c-KIT*, V-KIT Hardy-Zuckerman 4 feline sarcoma viral oncogene homolog; CML, chronic myeloid leukaemia; ET, essential thrombocythemia; *EZH2*, enhancer of zeste, drosophila, homolog 2; F, female; HCT, hematopoietic stem cell transplantation; *KRAS*, V-KI-RAS2 Kirsten rat sarcoma viral oncogene homolog; M, male; MDS, myelodysplastic syndrome; MF, myelofibrosis; MDS, myelodysplastic syndrome; µL, microliter; MPN, myeloproliferative neoplasm; MPN-U, myeloproliferative neoplasm unclassifiable; Nb., number; ng/mL, nanogram per millilitre; NGS, next-generation sequencing; *NPM1*, nucleophosmin 1; *NRAS*, neuroblastoma rat sarcoma viral oncogene homolog; PEG, percutaneous endoscopic gastrostomy; PMF, primary myelofibrosis; PV, polycythaemia vera; RBC, red blood cell count; *RUNX1*, runt-related transcription factor 1; SM-AHN, systemic mastocytosis with an associated haematological neoplasm; TD, transfusion dependent; TN, triple negative; y, years.

### 3.1. Molecular Testing for BCR-ABL1 and Classical Karyotyping Are Essential, Cost-Effective Elements in the Diagnostic-Work-Up of Myeloid Malignancies

Patient #1: A 65-year-old woman was referred as a TN prefibrotic myelofibrosis refractory to hydroxyurea, which she received for the last 14 months. Clinically, there was marked thrombocytosis with leukopenia and a left shift. Testing for *BCR-ABL1* and classical karyotyping were not done. *BCR-ABL1*-positive CML could be diagnosed through a positive PCR for *BCR-ABL1* combined with the detection of t(9;22)(q34.1;q11.2) by classical karyotyping. Evaluation of the BM confirmed the chronic phase of the disease. Therapy with the tyrosine-kinase inhibitor (TKI) nilotinib was started, and the patient achieved a hematologic, cytogenetic, and eventually a major molecular remission.

Patient #2: A 66-year-old female with transfusion-dependent (RBC-TD) anaemia and marked thrombocytosis. She was diagnosed as TN-ET and treated with hydroxyurea and anagrelide for the last 16 months. Reassessment revealed the lack of classical karyotyping. A bone-marrow examination revealed large and pleomorphic megakaryocytes with numerous mononuclear marginal and single lobed nuclei. Classical cytogenetics revealed a 5q-minus in 19 of 22 analysed metaphases.

The diagnosis of a 5q-minus syndrome was made, and therapy with lenalidomide was initiated. Normalization of platelet and haemoglobin values along with a complete cytogenetic remission was achieved.

### 3.2. Awarness of Systemic Mastocytosis as an Underdiagnosed Entity Must Be Upsurged

Patient #3: A 63-year-old male was referred with a diagnosis of TN-prefibrotic myelofibrosis. The patient suffered from a high disease burden (sweating, 20 kg weight loss, and

fatigue). In addition to massive splenomegaly, marked ascites and RBC-TD anaemia were present. Three BM biopsies over 12 months were performed.

Re-evaluation revealed a serum tryptase >200 µg/L (reference range <11 µg/L). BM examination showed a hypercellular marrow with delayed myelopoietic maturation, a normal blast count, and minimal reticular fibrosis. Multifocal clustering of >15 mast cells (MCs) co-expressing CD2, CD25, CD117, and mast cell tryptase were detected. BM cytology could not be performed because of dry tap. An activating mutation *KIT*D816V was present. A diagnosis of SM was made. Due to the presence of C-findings (RBC-TD anaemia and ascites), the criteria for aggressive SM (ASM) were fulfilled [8]. Treatment with the TKI midostaurin was started. Clinical improvement was fast and striking. After six months, the patient received a planned allogeneic hematopoietic cell transplantation (HCT).

Patient #4: This severely ill 55-year-old female referred with a diagnosis of MPN-U was previously published [9]. Again, the first clue to ASM was an elevated serum tryptase level of 49 ng/mL. Mast cells typical for mastocytosis were seen in the BM re-evaluation at our institute The subsequent molecular analysis revealed a classical activating p.D816V point mutation of the *cKIT*-gene and a mutated *KRAS*-gene. Treatment with midostaurin lead to a dramatic clinical improvement within three months.

Patient #5: A 52-year-old female was seen with a diagnosis of TN-PV with progressive leucocytosis and splenomegaly. Treatment consisted of phlebotomy, acetylsalicylic acid, and interferon alpha. Progression to AML with *NPM1* and *CDKN2A* mutations occurred a few months later, and the patient was referred to us. The serum tryptase level was >200 ng/mL. BM revealed a diagnosis of SM with an associated haematological neoplasm (SM-AHN). An activating point mutation in *cKIT* could not be detected. Induction chemotherapy with daunorubicin and cytarabine yielded blast clearance but persistence of MC aggregates. Therapy with the TKI Imatinib as bridging to HCT was initiated. Hepatosplenomegaly resolved, and HCT was performed. The patient is in complete remission since then.

Patient #6: A 63-year-old male presented with a diagnosis of MDS/MPN overlap. Clinically, pancytopenia and marked hepatosplenomegaly were present. The serum tryptase was 34 ng/mL. BM aspiration was a dry tap. BM biopsy showed a dysplastic haematopoiesis with 7% blasts in the bone marrow along with morphological criteria of SM. A *cKIT*D816V could not be detected. The diagnosis of SM-AHN (MDS with excess blasts-1 (MDS-EB1)) was made. Additionally, *EZH2* and *IDH2* mutations were found by NGS. The TKI midostaurin was initiated followed by HCT.

Patient #7: A 69-year-old female with a diagnosis of MPN-U was referred to us with a history of multiple liver abscesses caused by a Listeria monocytogenes infection. She was severely ill and complained of severe bone pain and weight loss. Clinically, there were pleural effusions and splenomegaly. Osteolytic bone lesions were detected by CT scan. Again, serum tryptase was >200 ng/mL. Classical cytogenetics revealed a monosomy 7, and NGS detected a mutation in the *NRAS* gene. *cKIT*D816V was absent. Again, BM aspiration was a dry tap. BM histological assessment revealed atypical mast cell infiltrates and MPN features enabling a diagnosis of SM-AHN, with MPN-U being the associated hematologic malignancy. The initiated therapy with imatinib had to be switched to midostaurin because of lack of efficacy. Azacidine was later added to control the progressive increase in blasts. Allogeneic HCT could not be performed because of ill health, and the patient eventually died.

*3.3. Benign Acquired and Hereditary Disorders Could Be the Cause of Reactive Bone Marrow and/or Peripheral Blood Abnormalities*

Patient #8: This 31-year-old patient referred with a diagnosis of TN-PV was previously published [10]. In addition to the unlikely diagnosis of TN-PV, the young age of the patient and a positive family history of polyglobulia were hints for a possible hereditary cause of erythrocytosis. Indeed, genetic work-up revealed the presence of a rare heterozygous point mutation in the beta-globin-chain (exon 2 (c.119A > C) leading to a change in codon 40 (CAG > CCG)). This variant belongs to the high-oxygen-affinity hemoglobinopathies.

Patient #9: A 75-year-old female with TN-triple-negative ET was seen. RBC-TD anaemia and refractory thrombocytosis (>3 million per μL) were predominant. The patient was treated with hydroxyurea, anagrelide, and cytarabine. When first seen, the patient described seasonal symptoms suggestive of mild Raynaud's phenomenon. A monoclonal serum immunoglobulin was not identified by protein electrophoresis, but an active autoimmune haemolytic anaemia, which was complement mediated by IgM cold agglutinins, was detected. Re-evaluation of the BM showed hyperplastic erythropoiesis that was initially misinterpreted as ET. There was no evidence of an underlying disease, such as aggressive lymphoma, other overt malignancies, or specific infections. Panel sequencing presented no mutations, and B-cell clonality analysis showed a physiological pattern. A diagnosis of primary chronic cold agglutinin disease (CAD) was made, and the patient was started on 50 mg prednisolone per day. Although only occasional patients are reported to respond to steroids [11], the response in the patient was dramatic, with normalization of haemoglobin and platelets within four weeks. Rituximab was given later to maintain the response and taper off steroids.

*3.4. Deep Sequencing Is Able to Detect Non-Canonical Somatic or Germline JAK2 or MPL Mutations in TN-MPN Patients*

Patient #10: A 56-year-old male was diagnosed with TN-PMF. Clinically, the patient presented with pronounced wasting, RBC-TD anaemia, and splenomegaly. Among others, a missense germline atypical *JAK2*-mutation in exon 25 (c.3323A > G; p.N1108S) and a somatic *ASXL1* mutation were detected [12]. Treatment with the JAK inhibitor ruxolitinib was started. After achieving a symptomatic response, an allogeneic HCT was conducted six months later.

Patient #11: A 41-year-old female presented with TN-MPN-U. Clinically, she had severe constitutional symptoms, such as fatigue as well as splenomegaly. Through a critical re-evaluation of the BM histology, a diagnosis of PMF was made; deep sequencing by NGS revealed an atypical mutation in exon 25 of the *JAK2* gene (c.3188G > A, p.R1063H). Therapy with ruxolitinib led to a fast and durable clinical response.

**4. Discussion/Conclusions**

As proposed by the WHO, the clinical cases underscore the importance of a multidisciplinary diagnostic approach that integrates sound clinical judgment, BM aspiration (cytology and karyotyping), and biopsy as well as molecular genetic features as partners of equal weight. This is the most suitable attempt to reach accurate diagnoses and to define disease entities, particularly in clonally ill-defined TN-MPN or MPN-U.

The identification of distinct, disease-driving mutations within the *JAK2*, *CALR*, or *MPL* genes has revolutionized the diagnostic and management landscape of classical *BCR-ABL1*-negative MPN. Deep sequencing allows the search for other non-driver "most frequent" mutations, e.g., in *ASXL1*, *EZH2*, *TET2*, *IDH1/IDH2*, *SRSF2*, and *SF3B1* [13,14].

Further, whole or targeted exome-sequencing approaches in TN-MPN have identified numerous novel somatic or germline mutations that occur in alternative exons of both *JAK2* and *MPL*. Many of these mutations result in a gain of function by inducing ligand-independent JAK2-STAT5 signalling [5,6]. Such non-canonical mutations could even be detected in around 10% of patients with classical *JAK2*V617F and *MPL*W515L-positive myelofibrosis [12].

Yet, in this illuminating era of expanding and extensive molecular diagnostic techniques, it is of utmost importance not to ignore or overlook well-established standard karyotyping and simple molecular testing for *BCR-ABL1* in patients labelled with a diagnosis of TN-MPN because of the immeasurable diagnostic, prognostic, and therapeutic consequences, as was the case in patients #1 and #2.

At this point, it must be emphasized that phenotype-driver mutations, such as JAK2, CALR, and MPL mutations, as well as the other non-driver "recurrent" mutations are not mutually exclusive and not exclusive to a particular MPN. This is most likely also true

for non-canonical *JAK2* and *MPL* mutations. A positive mutation assay establishes the presence of a clonal disorder but not its identity, as was the case in patients #10 and #11.

In all instances, sufficient BM aspiration should be secured in order to allow for an appropriate cytological evaluation and screening for driver and other mutations as well as classical cytogenetic analysis. The continual challenge remains in how to incorporate the molecular discoveries into the ever-evolving MPN molecular diagnostic algorithm.

With regard to BM findings, particular attention should be paid to BM morphology, as morphological features remain a central distinguishing feature in the 2016 edition of WHO classification system [1]. Yet, controversies concerning the BM morphology of different MPN subtypes is ongoing [2]. Although the overall BM evaluation shows a high degree of reproducibility, the identification of rare but specific morphological features displays a more limited reproducibility among different haemato-pathologists [4,15–17]. Further, poor-quality BM specimens, staining artefacts, treatment-induced BM modifications, and less experienced examiners might, in part, account for these shortcomings.

Certainly, advancements in the characterization and standardization of morphological BM features yielded an improvement in the differentiation of MPN subtypes, and educational workshops for haemato-pathologists can improve the integration of all histological characteristics into meaningful, reproducible subtyping of MPNs. Integrating cytological findings from a BM aspiration with BM biopsy histology is of utmost importance. Further, providing clinical data to the pathologist is an effective and simple approach to an improved diagnosis. Indeed, the histologic consensus of 53% among haemato-pathologists when BM evaluation was performed blinded to all clinical data increased to 83% when clinical data were taken into consideration with a concordance of 71% with the clinician's diagnoses [15].

Thus, a multidisciplinary discussion between clinicians, haemato-, and molecular-pathologists is the most powerful key to establishing a valid diagnosis in daily practice.

This is particularly true for patients with SM, which is, unfortunately, often overseen by both clinicians and less experienced pathologists. This was the case in patients #3 to #7. In fact, SM is the most frequent diagnostic revision we make in "real-world" practice in patients referred to the University Hospital Halle because of treatment-refractory TN-MPN or ill-defined MPN-U. For a phenotypic heterogenous disorder like SM—a true chameleon of internal medicine—routine screening with serum tryptase, although only a minor diagnostic criterion, is a simple and cheap clue for an eventual underlying SM and increases the awareness of physicians and pathologists to this disorder. In contrast to routine molecular screening for a *KIT* mutation, which is absent in 15% of patients with SM—as was the case in patients #5 to 7—serum tryptase is elevated in the majority of patients with SM [8].

Finally, benign acquired and hereditary non-clonal disorders as a possible cause of reactive bone marrow and/or peripheral blood abnormalities need to be excluded. Scanty information is available regarding the wide spectrum of reactive BM abnormalities which could be seen in the "real-world" setting. Benign hereditary or acquired non-clonal conditions might imitate MPN phenotypically, and the reactive BM abnormalities might be misinterpreted by less experienced examiners as surrogate features for MPN. Frequently, a thorough history and clinical examination provide helpful hints for an underlying benign disease and guides the further diagnostic work-up, as was the case in patients #8 and #9. Of note is that for some patients with TN-MPN, particularly TN-ET, there is no evidence of clonal haematopoiesis by X chromosome inactivation patterns. Whether these cases truly have a clonal malignancy is also questionable, as a polyclonal haematopoiesis is a feature consistent with a hereditary disorder [5,6]. In this work, we tried to highlight some aspects that accompany being faced with seemingly complex cases of MPN in everyday clinical practice, such as the importance of multidisciplinary, the fact that a correct diagnosis does not always require sophisticated and expensive techniques, and that the clue might be found in a thorough history, clinical examination, or simple and available tests, such as

that for *BCR-ABL1* and classical karyotyping. In addition, the awareness of the possibility of an underlying rare disease, such as SM, could be of value.

Yet, the report reflects a monocentric experience, and we could not claim to be exhaustive in the listing of possible pitfalls. The actual incidence of other underlying disorders in ill-defined MPNs or MDS/MPN overlap could only be answered in a prospective, well-designed, multicentre trial. In addition to the diagnostic endpoints of such a trial, scientific programs such as the analysis of bone-niche dysregulations and the identification of a possible cross talk between bone niche and immune system, which may contribute to propagating disease progression and mediating drug resistance, could be pursued [18,19].

In conclusion, an accurate diagnosis through distinguishing between the different subtypes of MPN and their separation from MDS/MPN overlap and SM is of utmost clinical importance, as treatment options and outcome for the different subtypes vary significantly. In clonally ill-defined TN-MPN or MPN-U, a multidisciplinary re-evaluation where molecular genetic features and BM findings along with a sound clinical judgment are integrated as partners of equal weight is crucial.

**Author Contributions:** Conceptualization, S.S., N.J. and H.K.A.-A.; methodology, all authors; validation, A.H., M.B. and C.W.; formal analysis, all authors.; investigation, all authors; data curation, all authors; writing—original draft preparation, S.S. and H.K.A.-A.; writing—review and editing, all authors; visualization, all authors; supervision, H.K.A.-A., C.W.; project administration, S.S. and C.L.H.N.; funding acquisition, not applicable. All authors have read and agreed to the published version of the manuscript.

**Funding:** This research received no external funding.

**Institutional Review Board Statement:** Ethical review and approval were waived for this study. This is a summary of clinical cases from routine care. A clinical trial regarding diagostics or therapy did not take place. Patients were diagnosed and treated according to current standards using approved methods.

**Informed Consent Statement:** Informed consent was obtained from all subjects involved in the study.

**Data Availability Statement:** Not applicable.

**Conflicts of Interest:** The authors declare no conflict of interest.

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
