# Peer review of "Clinical Discernment, Bone Marrow, and Molecular Diagnostics Are Equally Important to Solve the Phenotypic Mimicry among Subtypes of Myeloproliferative Neoplasms"

_reports, doi:10.3390/reports4030027_

Round 1

Reviewer 1 Report

Susann Schulze et al. uncovered multidisciplinary reevaluation where molecular genetic features, BM histology along with clinical judgment as key elements in Myeloproliferative Neoplasms. Point to be considered:

  1. I would slightly restructure the manuscript as follow: 

    First: Select a case and identify the message you wish to communicate, as well as your audience. Is this case report about an outcome, a diagnostic assessment, an intervention, a new or rare disease, or something else?

    Second: Gather the necessary information to accurately write WHAT happened as a timeline and as a narrative. Create the timeline of your case report—a visual summary of WHAT happened in the case report (see examples of timelines that follow the CARE guidelines) before writing the narrative section.

    Third: Complete the remainder of the case report using specialty-specific information if necessary, with appropriate scientific references and explanations. Support WHY an outcome occurred with reference to the scientific and historic literature whenever possible. Write the abstract last.

    De-Identification: Patient information must be de-identified and informed consent obtained prior to submitting your case report to a journal.

    Writing Sequence

    Part 1 — Working Title, WHAT happened: Timeline and Narrative

    Develop a descriptive and succinct working title that describes the phenomenon of greatest interest (symptom, diagnostic test, diagnosis, intervention, outcome).

    WHAT happened. Gather the clinical information associated with patient visits, in this case, report to create a timeline as a figure or table. The timeline is a chronological summary of the visits that make of the episodes of care from this case report.

    Narrative of the episode of care (including tables and figures as needed).

    The presenting concerns (chief complaints) and relevant demographic information.

    Clinical findings: describe the relevant past medical history, pertinent co-morbidities, and important physical examination (PE) findings.

    Diagnostic assessments: discuss diagnostic testing and results, a differential diagnosis, and the diagnosis.

    Therapeutic interventions: describe the types of intervention (pharmacologic, surgical, preventive, lifestyle) and how the interventions were administered (dosage, strength, duration, and frequency). Tables or figures may be useful.

    Follow-up and outcomes: describe the clinical course of the episode of care during follow-up visits including (1) intervention modification, interruption, or discontinuation; (2) intervention adherence and how this was assessed; and (3) adverse effects or unanticipated events. Regular patient report outcome measurement surveys  may be helpful.

    Part 2 — WHY it might have happened: Introduction, Discussion, Conclusion

    The introduction should briefly summarize why this case report is important and the other discussed tips can also be synthesized or enclosed in supplements but should be available

    WHY it might have happened. The discussion describes case management, including strengths and limitations with scientific references.

    The conclusion, usually one paragraph, offers the most important findings from the case without references.

    Part 3 — Abstract, Key Words, References, Acknowledgement, and Informed Consent

    Abstract. Briefly summarize in a structured or unstructured format the relevant information without citations. Do this after writing the case report. Information should include (1) Background, (2) Key points from the case; and (3) Main lessons to be learned from this case report.

    Informed Consent and Patient Perspective. The patient should provide informed consent (including a patient perspective) and the author should provide this information if requested. Some journals have consent forms that must be used regardless of the informed consent you have obtained.  It is often best to ask for informed consent and the patient’s perspective before you begin writing your case report/case series.

    2. The limitations and strengths of this report should be better evaluated

    3. This reviewer personally misses some important consequences that the authors' findings can bring to the scientific community while discussing their findings from an alternative standpoint: a correlation between bone niche and cancer cells in AML has been described. In this case, osteoblasts promote the progression and transformation of the myeloid cells lineage in preneoplastic and neoplastic cells. Specifically, the osteoblasts are able to slow down leukemia progression through an unfavorable microenvironment for leukemic blast growth. The “bone niche” concept becomes a “niche-induced leukemia” system: for the first time bone niche is evaluated as a dynamic system that includes bone, immune and cancer cells. The underlying message here is that more precision and individualized approaches need to be tested in well-designed clinical trials – a challenge, but I would be interested in their perspective of how this might be done (please refer to PMID: 32064051).

Author Response

“Clinical Discernment, Bone Marrow, and Molecular Diagnostics are Equally Important to Solve the Phenotypic Mimicry Among Subtypes of Myeloproliferative Neoplasms“

Manuscript: 1318118

Dear Reviewer,

thank you very much for reviweing our manuscript. We would like to resubmit a revised version In this covering letter we would list all amendments and responses to the points raised by the reviewers and how we have dealt with them in the manuscript.

Reviewer 1

Point 1: I would slightly restructure the manuscript  

Response 1: thank you very much for the advice regarding the guidelines published under „How to Write a Case Report — CARE Case Report Guidelines (care-statement.org)“

These guidelines are indeed helpful when writing A detailed Case Report. They were followed by us when we published two of the patients included in this series (reference 9 and 10). Yet, in this report we used the collection of 11 real-world patients to demonstrate clinical diagnostic challenges in everyday practice which we gathered under clear headings with a clear message. Each individual patient could not be detailed in more length because of space limitations and redundancy. Nevertheless, the main and relevant data to history, clinical examination, laboratory, bone marrow and molecular findings were systematically mentioned for each patient.

Point 2: The limitations and strengths of this report should be better evaluated

Response 2: we have added the following at the end of the discussion:

„In this work, we tried to highlight some aspects when faced with seemingly complex cases of MPN in everyday clinical practice such as the importance of multidisciplinary, the fact that a correct diagnosis does not always require sophisticated and expensive techniques, and that the clue might be found in a thorough history, clinical examination, or simple and available tests such as that for BCR-ABL1 and classical karyotyping. Also the awareness of the possibility of an underlying rare disease such as SM could be of value. Yet, the report reflects a monocentric experience and we could not claim to be exhaustive in the listing of possible “pitfalls”. The actual incidence of other underlying disorders in ill-defined MPNs or MDS/MPN overlap could only be answered in a prospective well-designed multicenter trial.“

Point 3: This reviewer personally misses some important consequences that the authors' findings can bring to the scientific community while discussing their findings from an alternative standpoint: a correlation between bone niche and cancer cells in AML has been described. In this case, osteoblasts promote the progression and transformation of the myeloid cells lineage in preneoplastic and neoplastic cells. Specifically, the osteoblasts are able to slow down leukemia progression through an unfavorable microenvironment for leukemic blast growth. The “bone niche” concept becomes a “niche-induced leukemia” system: for the first time bone niche is evaluated as a dynamic system that includes bone, immune and cancer cells. The underlying message here is that more precision and individualized approaches need to be tested in well-designed clinical trials – a challenge, but I would be interested in their perspective of how this might be done (please refer to PMID: 32064051).

Response 3: we wrote this report from the perspective of a diagnostic work-up in daily practice as missing the right diagnosis has such grave consequences on the outcome of patients. Nevertheless and indeed, scientific questions are extremely interesting. Particularly the mentioned osteoimmunology knowing that the bone niche is strongely dysregulated in patients with MPN as was reviewed recently by Curto-Garcia N, Harrison C, McLornan DP. Bone marrow niche dysregulation in myeloproliferative neoplasms. Haematologica. 2020, 105, 1189–1200.

To elaborate on scientific questions in this context, we added the following to the under point 2 mentioned new text the following:

“In addition to the diagnostic endpoints of such a trial, scientific programs such as the the analysis of bone niche dysregulations and the identification of a possible cross talk between bone niche and immune system which may contribute to propagating disease progression and mediating drug resistance could be pursued [18,19].”

The two new following references (18 and 19) were added:

  • Curto-Garcia N, Harrison C, McLornan DP. Bone marrow niche dysregulation in myeloproliferative neoplasms. Haematologica. 2020, 105, 1189–1200.
  • Antonio G, Oronzo B, Vito L, Angela C, Antonella A, Roberto C, Giovanni SA, Antonella L. mmune system and bone microenvironment: rationale for targeted cancer therapies. Oncotargets 2020, 11:480-487.

We hope that we have adequately answered the points raised by the reviewer

Sincerely

Yours

Susann Schulze and coauthors

Reviewer 2 Report

  1. General.

The article is well written and the authors do indeed meet the goal as set out in the title.

 However, the article does not mention the importance of bone marrow cytology as a complement to bone marrow histology. In some cases, BM aspirate cytology might be more suitable than BM histology. For instance,  it rouses suspiscion of mastocytosis earlier and more clearly than BM histology. A multidisciplinary reevaluation should therefore not only include bone marrow histology, but also bone marrow aspirate cytology (anatomical pathologist and clinical pathologist).

  1. Patients and Methods.                                                                           Please specify whether the bone marrow biopsies and smears that were re-evaluated were fresh samples obtained after patient referral to your institution, or otherwise.                                                                          
  2. Results.

Modify the sentence: “clustered into four categories (A to D) based on diagnostic entities and how the diagnoses were made” to “clustered into four categories (A to D) based on diagnostic entities and/or how the diagnoses were made”.

Label the four clusters A to D (sub headings 3.1. -3.4).

Clarify whether the somatic gene mutation was identified prior to or at re-evaluation for patients #4 and #5, as it’s not clear from the text.

  1. Discussion.

The following sentence (page 9, lines 6 -8) is ambiguous: “A positive mutation assay establishes the presence of a hemapoietic stem-cell disorder...” Why “hemapoietic stem-cell disorder” and not clonal disorder?

Please clarify what is meant by the phrase  (page 9, lines 21-22)“Additionally, it needs to be remembered that BM histologic features are surrogate markers and cannot provide specificity.” Surrogate markers for what? For NGS? Are you suggesting that NGS could replace BM histology? According to the WHO criteria, diagnosis of MPN is based to this day on BM morphology which remains fundamental to its diagnosis.

Author Response

“Clinical Discernment, Bone Marrow, and Molecular Diagnostics are Equally Important to Solve the Phenotypic Mimicry Among Subtypes of Myeloproliferative Neoplasms“

Manuscript: 1318118

Dear Reviewer,

thank you very much for reviweing our manuscript. We would like to resubmit a revised version In this covering letter we would list all amendments and responses to the points raised by the reviewers and how we have dealt with them in the manuscript.

Reviewer 2

Point 1: General.

The article is well written and the authors do indeed meet the goal as set out in the title. However, the article does not mention the importance of bone marrow cytology as a complement to bone marrow histology. In some cases, BM aspirate cytology might be more suitable than BM histology. For instance,  it rouses suspiscion of mastocytosis earlier and more clearly than BM histology. A multidisciplinary reevaluation should therefore not only include bone marrow histology, but also bone marrow aspirate cytology (anatomical pathologist and clinical pathologist).

Response 1: Indeed, bone marrow cytology is of utmost importance. Although, bone marrow aspiration is mentioned in the text as follows: „it needs to be remembered that the diagnosis of MPN is a multidisciplinary task requiring consideration of the presenting clinical features, morphological assessment of the peripheral blood and bone marrow aspirate and biopsy“, we have now emphasised throughout the manuscript the fact that bone marrow (including both cytology and histology) is needed. In the title Bone Marrow „without histology“ appears and the wording of bone marrow examination is clarified to stress that both cytology and histology are fundamental.

To further stress this, the following sentence was included in the discussion: „Integrating cytological findings from a BM aspiration with BM biopsy histology is of utmost importance.“

In our institution cytology is evaluated by ourselves (the hematologists).

  1. Patients and Methods.

Point 2: Please specify whether the bone marrow biopsies and smears that were re-evaluated were fresh samples obtained after patient referral to your institution, or otherwise.

Response 2: The bone marrow biopsies and smears were fresh samples obtained after patient referral to our institution. This has been added for clarification.

  1. Results.

Point 3: Modify the sentence: “clustered into four categories (A to D) based on diagnostic entities and how the diagnoses were made” to “clustered into four categories (A to D) based on diagnostic entities and/or how the diagnoses were made”.

Response 3: The sentence was adjusted accordingly.

Point 4: Label the four clusters A to D (sub headings 3.1. -3.4).

Response 4: The adjustment has been made accordingly.

Point 5: Clarify whether the somatic gene mutation was identified prior to or at re-evaluation for patients #4 and #5, as it’s not clear from the text.

Response 5: In both patients, somatic mutations were dedected during reevaluation. For clarity, the sentence now reads: „The subsequent molecular analysis revealed a classical activating p.D816V point mutation of the cKIT-gene and a mutated KRAS-gene.”

  1. Discussion.

Point 6: The following sentence (page 9, lines 6 -8) is ambiguous: “A positive mutation assay establishes the presence of a hemapoietic stem-cell disorder...” Why “hemapoietic stem-cell disorder” and not clonal disorder?

Response 6: The sentence has been changed to clonal disorder

Point 7: Please clarify what is meant by the phrase  (page 9, lines 21-22) Surrogate markers for what? For NGS? Are you suggesting that NGS could replace BM histology? According to the WHO criteria, diagnosis of MPN is based to this day on BM morphology which remains fundamental to its diagnosis.

Response 7: we have stressed thoughout the manuscript that NGS and bone marrow are complementary and neither could be substituted by the other. To avoid confusion, we omitted the sentence: “Additionally, it needs to be remembered that BM histologic features are surrogate markers and cannot provide specificity.”

We hope that we have adequately answered the points raised by the reviewer

Sincerely

Yours

Susann Schulze and coauthors

Round 2

Reviewer 1 Report

The authors  addressed all issues.